# Language-Conditioned Reinforcement Learning to Solve Misunderstandings with Action Corrections

**Frank Röder**[*]
Institute for Data Science Foundations
Hamburg University of Technology
`frank.roeder@tuhh.de`

**Manfred Eppe**
Institute for Data Science Foundations
Hamburg University of Technology
`manfred.eppe@tuhh.de`

## Abstract

Human-to-human conversation is not just talking and listening. It is an incremental process where participants continually establish a common understanding to rule out misunderstandings. Current language understanding methods for intelligent robots do not consider this. There exist numerous approaches considering non-understandings, but they ignore the incremental process of resolving misunderstandings. In this article, we present a first formalization and experimental validation of incremental action-repair for robotic instruction-following based on reinforcement learning. To evaluate our approach, we propose a collection of benchmark environments for action correction in language-conditioned reinforcement learning, utilizing a synthetic instructor to generate language goals and their corresponding corrections. We show that a reinforcement learning agent can successfully learn to understand incremental corrections of misunderstood instructions.

## 1 Introduction

In real-world human-to-human communications, misunderstandings happen very frequently. Ambiguity, a lack of common ground, and incorrect statements are three examples of situations that cause misunderstandings. For example, you may ask your colleague to "give me the item on the right", referring to the object on *her* right, while your conversation partner assumes you refer to *your* right. Humans are extremely proficient in resolving such ambiguities by performing conversational repair, e.g., "No, not to my right, I mean your right!". Such repair commands enable conversation partners to incrementally establish a common understanding of what was said.

The state-of-the-art in robotic language understanding considers non-understandings [Bohus and Rudnicky, 2008, Bordes et al., 2017], but it entirely ignores the incremental resolution of misunderstandings with conversational repair.

The concept of action correction is not only important when the system is being deployed into the real world but should be, as we propose in this article, a fundamental part of the learning procedure. However, even flagship approaches like SayCan struggle with ambiguities and negations [Ahn et al., 2022], despite using large language models with a rich semantic knowledge of the world.

Recent advances in controlling robots with language achieved remarkable results in simulation [Lynch and Sermanet, 2021, Huang et al., 2022] and also made considerable progress in real-world applications [Shridhar et al., 2021]. Especially reinforcement learning (RL) turns out to be a suitable framework for training robots paired with large language models [Ahn et al., 2022]. But it remains unclear how we can build an intelligent robot capable of resolving ambiguities, as we depict in Figure 1.

In this article, we propose an extension of goal-conditioned RL, where goals are specified as word

---

[*]Corresponding author (`https://www.dsf.tuhh.de/index.php/team/frankroeder/`)

Accepted to the $2^{nd}$ Workshop on Language in Reinforcement Learning, (NeurIPS 2022). Do not distribute.

embeddings that can be dynamically extended with repair commands while a robot is executing a certain task or after causing changes in the environment (see Figure 1). We hypothesize that our approach enables robotic agents to react appropriately to the following three yet unexplored cases of misunderstandings (see Appendix A.3 for examples in our environment):

**Ambiguity**   Instructions are ambiguous if they are underspecified. As an example, consider Figure 1. Here, the instruction *"grasp the cube"* is imprecise as there are two cubes and the instruction is referring to a categorical feature of the present objects that are identical for two entities.

**Lack of common ground**   When operator and robot have a lack of common ground, the robot misunderstands the instruction due to unknown words used (out-of-distribution) or insufficient training. For example, if the robot's object detection method misclassifies a red apple for a tomato, the operator needs to use action corrections to interrupt the robot's interactions with the apple and guide it to the tomato.

**Instruction Correction**   It can happen that the error is completely on the operator's side, e.g., when she/he accidentally utters an unintended instruction. For example, the operator might say *"reach the red cube"*, while having intended to instruct the robot to reach the green cube. This requires an action correction like "No, sorry, I mean the green cube".

In the following, we first highlight that the state of the art in robotic language understanding lacks methods to address such misunderstandings. Then we briefly describe our method, and, finally, we present our environment extension. We conclude the article with our experiments and results, showing how our method successfully fills this gap.

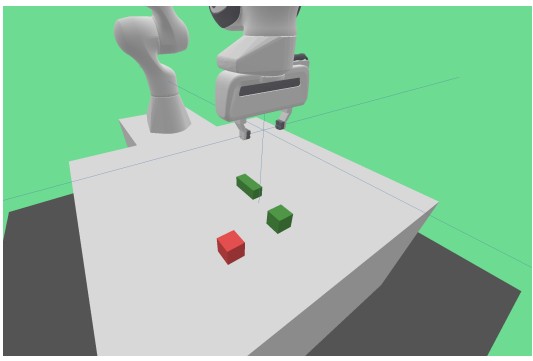

Figure 1: Consider the present scene with a green cube, a green cuboid, and a red cube. An instructor assigns the robot to *"grasp the cube"*. The robot might grasp an object by chance (let us assume the red cube). Now an action correction like *"actually, the green"* could resolve the robot's mistake caused by the ambiguity of the instruction. The robot should be still aware of the *"grasping"* task and initial shape word *"cube"*, to ideally ignore the green cuboid and approach the green cube.

## 2   Related Work

Natural language is the most important tool of communication, therefore researchers investigate its usage for robots for more than a decade [Steels, 2008] and it is still actively researched [Tellex et al., 2020]. Specifically grounded language is a key ingredient for true understanding because it is not always possible to transmit information without the background knowledge and considerations of other modalities [Bisk et al., 2020]. The notion of controlling intelligent agents with language has also found its way into deep RL research [Hill et al., 2019, Akakzia et al., 2021, Chevalier-Boisvert et al., 2019]. The review of Luketina et al. [2019] outlines the usage of language in RL where it is applied to assist or instruct an agent to solve a task. RL is an embodied setup and therefore provides the grounding of language through rewards [Hermann et al., 2017, Akakzia et al., 2021, Röder et al., 2022]. Furthermore, its compositional representation allows decomposing a task, e.g., *"remove the plate and clean the table"*, into two subtasks, namely *"remove plate"* and *"clean table"*. We assume that action corrections are of great benefit if such a compositional representation is utilized by the agent because a misunderstanding could originate in misinterpretation of one single word.

## 3 Method

We outline the general notion of action correction as dynamic goal extension, where we augment the episodic goal by the action correction. For a general example, consider the illustration in Figure 2. Here, the agent interacts with the world under the consideration of the instruction $g_\ell$ [2]. The instructor is observing the environment state for interactions with wrong objects, on which an action correction is returned immediately or, like in the real world, with a small delay, after some actions potentially already changed the world state $s_t$. The following goal is now a concatenation of the original goal and the action correction, $g_\ell \circ g_{ac}$. The fact that the policy $\pi(s_t, g_\ell)$ is conditioned on the current state $s_t$ and instruction $g_\ell$, requires it to notice the connection the between action correction $g_{ac}$, the initial state $s_t$ and the state $s_{t+1}$ the correction appears.

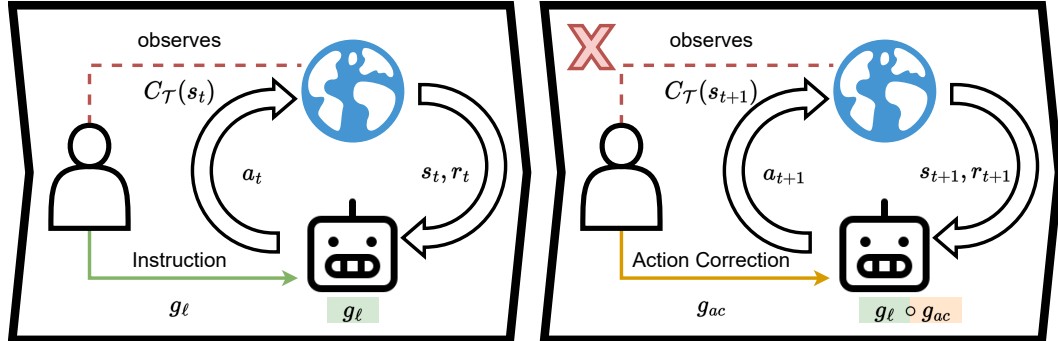

Figure 2: We illustrate the general case of an action correction. The episodic conversation starts with an instruction $g_\ell$ uttered by the instructor (left box). The agent, being in state $s_{t-1}$ executes an action $a_t$ to receive a scalar reward $r_t$ and the next world state $s_t$. However, the instructor detects with the help of the task-specific condition function from Equation 1 a wrong behavior. Following, an action correction $g_{ac}$ is returned, which is concatenated to the original goal $g_\ell \circ g_{ac}$ (right box). Under the consideration of the new linguistic goal, the agent attempts to correct its behavior.

Formally, the reward function $R_g$ is determined by at least one or more task-specific conditions $C_\mathcal{T}$ [Röder et al., 2022], that are evaluated at each step of the simulation (see Equation 1).

$$r_t = R_g(s_{t+1}, g_\ell) = \begin{cases} 0, & \text{if } C_\mathcal{T}(s_{t+1}, g_\ell) \\ -1, & \text{otherwise} \end{cases} \tag{1}$$

The synthetic instructor uses the same function $C_\mathcal{T}$ from Equation 1 to also detect interactions with the non-goal objects. In case of a misunderstanding, the instructor returns an action correction, while having the full knowledge of the goal object and non-goal object's position and properties in the scene. Figure 1 illustrates an example of an ambiguity where the robot interacts with the wrong object because of a spatial setup (grasping the closer object) or by chance. Besides, the correction in this example is referring to a unique property of the actual goal object and is not making use of negations like *"not the red"*. We are aware of a recent experiment with negations in language-conditioned RL [Hill et al., 2020], but the authors are not utilizing them for action correction. They rather instruct an agent to go to an object that has not a specific property. However, we assume this to be learned in our scenarios when the corrections involve negations. In the following section, we give a brief description of our environments for action correction.

### 3.1 Learning Environment for Action Correction

We present an extension of the publicly available language-conditioned RL environment LANRO [Röder et al., 2022] [3]. It provides a selection of 4 tasks, such as *reach*, *push*, *grasp*, and *lift*. An episode consists of two or three objects on a table (see Figure 1) and a linguistic goal generated by a synthetic instructor. We provide the formal definition of the templates to generate the instructions in Appendix A.1. Like Röder et al. [2022], we make use of a condition-based reward function corresponding to the language goal (see Equation 1). We use the task-specific conditions to determine

---

[2]We omit the subscript $t$ to emphasize the episodic language goal $g_\ell$, regardless of the time.

[3]https://github.com/frankroeder/lanro-gym

the moment we return an action correction for, e.g., an interaction with a wrong object (see Figure 2). From this point on, the original goal $g_\ell$ is extended by the action correction $g_{ac}$. In the end, we receive a concatenation of strings, hence $g_\ell \circ g_{ac}$. To bootstrap the learning of simple instruction-following, we challenge the agent to solve action corrections in only 50% of the episodes. Since we actually delegate the agent to fulfill the task of two different linguistic goals, the episode limit is increased to 100 steps. In summary, the action correction episodes contain at most two goals that need to be solved in sequence. The instructor can resolve ambiguous and erroneous statements immediately (longer initial instruction goal) or after recognizing the wrong behavior (extended goal given changed world state). Finally, the lack of common ground could be resolved by both repeating the goal object properties or providing a negation after interacting with a wrong object.

## 4   Experiments

For our initial experiments, we make use of a language-conditioned Soft Actor-Critic baseline [Akakzia et al., 2021, Röder et al., 2022], that we pair like Röder et al. [2022] with a GRU as language encoder [Cho et al., 2014]. We conduct all the experiments with two and three objects on the table. The objects are sampled uniformly from all attribute combinations, but we put specific constraints on the overlapping features. Our goal object has at most 1 property in common with the non-goal objects. This allows us to create scenarios like the one in Figure 1, where the agent needs to consider every part of the language goal. Furthermore, it assures that there is a unique and valid solution after the instructor proposes the correction.

In Figure 3, we show the mean and the standard deviation of 3 random seeds in the *"reach"* and *"push"* tasks. Next to the default action corrections without negations (AC, red lines in Figure 3), we showcase the episodes with action corrections including negations (ACN, purple lines in Figure 3). As we sample 50% of the episodes with an action correction challenge, we provide additional plots for those episodes in Appendix A.2.

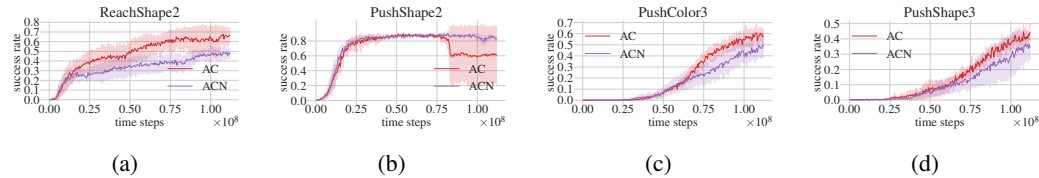

Figure 3: Our experiments showcase the success rate of a selection of *"reach"* and *"push"* experiments with different object configurations. On the x-axis, we show the total environment steps. The y-axis shows the mean success rate of solving both the episodes with single language goals and those including the action correction. In Appendix A.2, we depict the action correction success rates only.

As expected, learning action correction with negations is slightly more difficult than corrections created with editing terms and directly naming the desired goal object properties. Learning action corrections with just two objects in the scene is basically switching to the other object, after the action correction appears. However, it might still help with learning a robust representation because the agent nevertheless needs to grasp the nuances of the language. In Figure 3c and Figure 3d, the performance in the *"push"* task with 3 objects is shown. Although the success rate is increasing over time, the baseline requires millions of environment steps to surpass the threshold of only solving the single goal instructions shown in Figure 3c. We excluded the tasks that were too challenging for our baseline, but we expect addressing this with methods like hindsight learning (Appendix A.4), intrinsic motivation [Röder et al., 2020, Colas et al., 2020b, Akakzia et al., 2021] or hierarchical RL [Jiang et al., 2019, Eppe et al., 2022].

In summary, it still takes additional effort to solve action correction challenges efficiently. However, we postulate our environment and initial experimental results to be valuable insight for future work.

## 5   Conclusion

This paper is the first to formalize and experimentally validate the use of incremental action correction for robotic instruction-following based on reinforcement learning. We present a novel collection of benchmark environments for which we show initial experimental results, using a common language-conditioned baseline algorithm.

## Acknowledgments

Frank Röder and Manfred Eppe acknowledge funding by the DFG through the LeCAREbot (433323019) and IDEAS/MoReSpace (402776968) projects.

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

# A  Appendix

## A.1  Grammar Templates

### A.1.1  Instruction Templates

$$
\begin{aligned}
\langle\text{INSTRUCTION}\rangle &\models \langle\text{TASKVERB}\rangle\ \langle\text{ARTICLE}\rangle\ \langle\text{OBJECT}\rangle \\
\langle\text{OBJECT}\rangle &\models \langle\text{COLOR}\rangle\ \langle\text{SHAPE}\rangle\ |\ \langle\text{SHAPE}\rangle\ \texttt{object}\ |\ \langle\text{COLOR}\rangle\ \texttt{object} \\
\langle\text{SHAPE}\rangle &\models \langle\text{CUBE}\rangle\ |\ \langle\text{CUBOID}\rangle\ |\ \langle\text{RECTANGLE}\rangle \\
\langle\text{COLOR}\rangle &\models \texttt{red}\ |\ \texttt{green}\ |\ \texttt{blue}\ |\ \texttt{yellow}\ |\ \texttt{purple}\ |\ \texttt{orange}\ |\ \texttt{pink}\ |\ \texttt{cyan}\ |\ \texttt{brown} \\
\langle\text{TASKVERB}\rangle &\models (\texttt{reach}\ |\ \texttt{touch}\ |\ \texttt{contact})\ |\ (\texttt{push}\ |\ \texttt{move}\ |\ \texttt{shift}) \\
\langle\text{CUBE}\rangle &\models \texttt{cube}\ |\ \texttt{box}\ |\ \texttt{block} \\
\langle\text{CUBOID}\rangle &\models \texttt{cuboid}\ |\ \texttt{brick}\ |\ \texttt{oblong} \\
\langle\text{CYLINDER}\rangle &\models \texttt{cylinder}\ |\ \texttt{barrel}\ |\ \texttt{tophat} \\
\langle\text{ARTICLE}\rangle &\models \texttt{the}
\end{aligned}
$$

Figure 4: Backus normal form (BNF) for our instruction set inspired by Chevalier-Boisvert et al. [2019]. Here we show the verbs for the *"reach"* and *"push"* task only.

### A.1.2  Action Correction Templates

$$
\begin{aligned}
\langle\text{EXTENDED INSTRUCTION}\rangle &\models \langle\text{INSTRUCTION}\rangle\ \langle\text{CORRECTION}\rangle \\
\langle\text{CORRECTION}\rangle &\models \langle\text{BEGINNING}\rangle\ \langle\text{ARTICLE}\rangle\ \langle\text{OBJECT}\rangle \\
\langle\text{BEGINNING}\rangle &\models \langle\text{EXCUSE}\rangle\ |\ \langle\text{NEGATION}\rangle\ |\ \langle\text{EDIT}\rangle \\
\langle\text{EXCUSE}\rangle &\models \texttt{sorry}\ |\ \texttt{excuse me}\ |\ \texttt{no i meant}\ |\ \texttt{pardon} \\
\langle\text{NEGATION}\rangle &\models \texttt{not} \\
\langle\text{EDIT}\rangle &\models \texttt{actually}
\end{aligned}
$$

Figure 5: The extended BNF grammar for our action corrections paired with the primal instruction. These definitions apply to all tasks.

## A.2 Action Correction Successes

In our experimental design, we decide to sample 50% of the episodes with the need to solve a second goal, our action correction. We provide additional correction success rate plots to make clear how capable the agent is in solving the language-conditioned tasks augmented by a misunderstanding challenge.

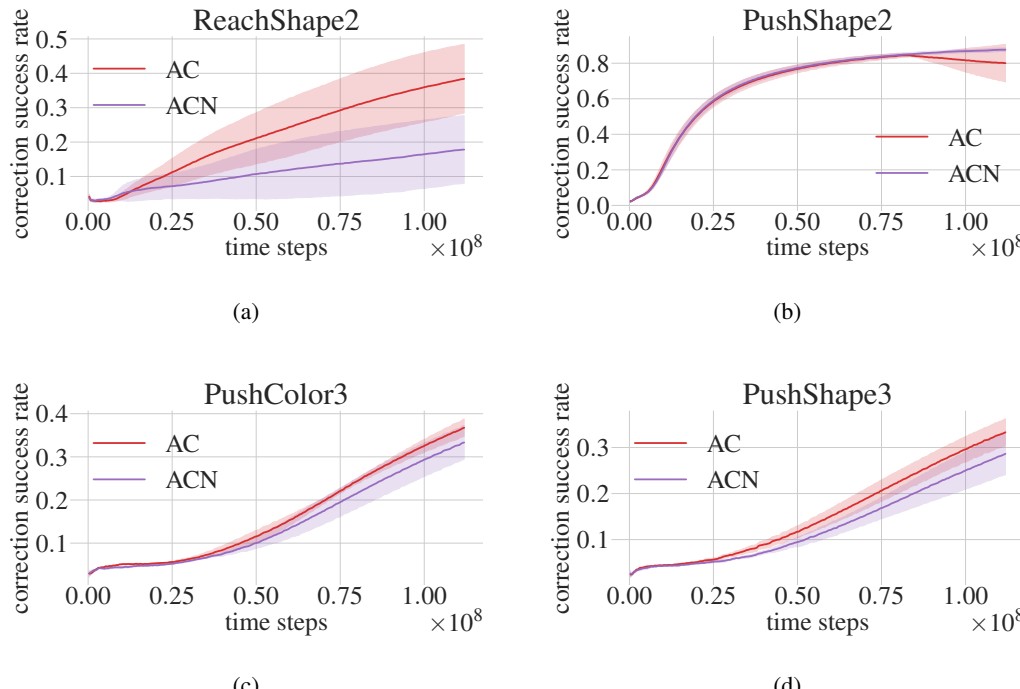

Figure 6: These figures depict the corresponding correction success rates of the experiment in Figure 3.

### A.3 Action Correction Example

This section showcases that all types of misunderstanding from Section 1 are covered by the proposed environment setup. In Figure 7, we present a selected scene for this purpose.

**Ambiguity**    For the case of an ambiguity, the operator instructs the robot to *"reach the cuboid"* (Figure 7a-c). However, given the two present cuboids, this instruction is ambiguous, and the robot might reach for the red cuboid (Figure 7d). As the instructor's intention was it to reach the blue cuboid, the ambiguous situation is resolved by the action correction *"sorry, the blue cuboid"*, after which the robot carries out the new goal successfully (Figure 7e-h). Instead of an action correction, we could also return an action correction with a negation like *"sorry, not the red one"*.

**Lack of common ground**    Consider the case where the instructor utters a rare word like *"azure"* in the instruction *"reach the azure cuboid"*. However, the robot still tries to make a guess and reaches for the closer cuboid (Figure 7a-c). Subsequently, the instructor can return an action correction that can resolve the uncertainty with, e.g. *"actually, the blue cuboid"*, or express additional information in the form of a negation like *"no, not the red"*.

**Instruction Correction**    For an instruction correction (where the operator's intention and instruction mismatches), an initial instruction like *"reach the red cuboid"* (Figure 7a-c) gets corrected after the instructor recognizes the wrong object being touched. Following, an action correction *"actually, the blue"* is returned (Figure 7d). Finally, the agent switches to the other object and finishes the task (Figure 7e-h).

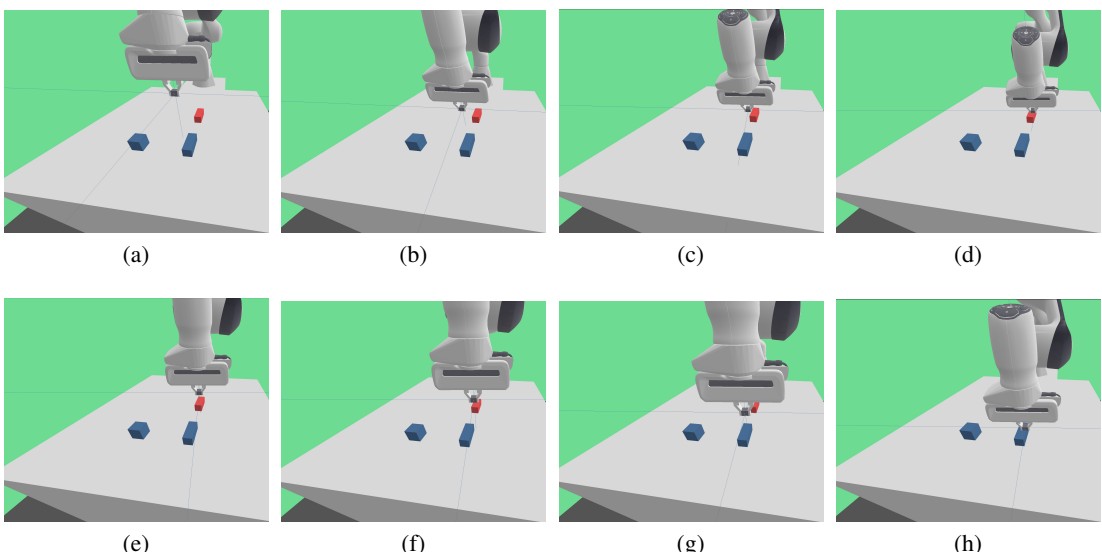

|     |     |     |     |
|:---:|:---:|:---:|:---:|
| (a) | (b) | (c) | (d) |
| (e) | (f) | (g) | (h) |

Figure 7: An action correction scene from LANRO [Röder et al., 2022]

## A.4 Hindsight Learning for Action Correction

We argue that language-conditioned RL has many properties with goal-conditioned RL in common and could benefit from its insights. In Colas et al. [2020a], the authors even speak for goal-agnostic agents where the goal originates in modalities like images, Cartesian Coordinates, language or other representations. The challenge of correcting a behavior is to interpret the action correction in context of the past actions, initial instruction and changes of the state. Because one can consider the initial instruction to be an abstract goal description, the action correction acquaints about an alternation of the episodic goal.

Especially methods of hindsight learning are valuable to apply to language-conditioned RL [Jiang et al., 2019, Colas et al., 2020b, Akakzia et al., 2021, Röder et al., 2022] because they help to improve the sample-efficiency by utilizing failures as imaginary successes for learning. For this article, we keep it a question for future work on how to combine this with action correction.

## A.5 Hyperparameters

The hyperparameters are a slight alternation of the ones used by Akakzia et al. [2021] and Röder et al. [2022].

Table 1

| Parameter. | Values. |
|---|---|
| batch size | 256 |
| hidden size of MLPs | 256 |
| learning rate actor, critic and entropy tuning | 1e$-$3 |
| number of workers | 16 |
| trade-off coefficient $\alpha$ | 0.2 |
| buffer size | 1e$+$6 |
| word embedding size | 32 |
| discount $\gamma$ | 0.98 |
| number of critics | 1 |
| polyak $\rho$ | 0.95 |

