# OpenReview forum: "Language-Conditioned Reinforcement Learning to Solve Misunderstandings with Action Corrections"
_NeurIPS.cc/2022/Workshop/LaReL — LaReL 2022_

### Official Review · Reviewer_bnn1 · 2022-10-05
**Interesting topic, very preliminary study**

**Rating:** 6
**Confidence:** 3

**Review:**

The authors are interested in letting tutors specify action corrections to instruction-following agents. They advocate for the need for such mechanisms in three contexts (when the initial goal specification is ambiguous, when the agent lacks a common ground with the tutor or when the instruction itself was wrong), they outline that such mechanisms are absent so far in the literature and they provide a preliminary study where they examine the success rates of instruction-following agents with a standard instruction and then with a instruction  + correction pair. They show that in the second case, learning is slower.

The identified domain is interesting, the study of the literature is relevant, the chosen experimental setup is fine, so I believe this is enough to let the authors present their work at the workshop.

But the empirical study is very preliminary. In the conclusion the authors claim that their work is the first to exprimentally validate the use of incremental action corrections, but I disagree with that. To me, the experiments only validate something already known, which is that training an instruction-following agent with a longer instruction is more difficult.

The experiments are missing several of the properties one could expect from an "action correction" context.

In particular, the authors are adding a second instruction 50% of time, but in a real "action correction" context the additional instruction should come while the agent is acting and the tutor is detecting that the agent is about to fail. This "online" correction process seems to be absent of the experiment, or at least is not described. Online failure detection and appropriate correction generation are not addressed either.

Besides, the authors identify three different contexts for "action correction", but it is unclear that there experiment truyl corresponds to one of these three contexts. A study of the different learning dynamics in the different contexts would be of interest.

Note that I would call "instruction correction" what the authors call "action correction", because the added instruction corrects the initial instruction rather than the action taken by the agent.

So I hope that before the time of the workshop, the authors will find the time to obtain more relevant results with respect to the problem they are willing to address.

Typos:

line 78, the dot should be at the end of the equation.

Subsection A.1, A.2 -> Appendix A.1, A.2...

In the caption of Fig. 3, no need to point to Fig. 3A, 3b, etc. Just mention (a,b,c,d) or even "top row, bottom row", we know we are in Figure 3.

---

### Official Review · Reviewer_G27j · 2022-10-17
**Relevant paper but experimental section needs to be further developed**

**Rating:** 6
**Confidence:** 3

**Review:**

This paper considers the task of action correction/repair for instruction-following tasks based on reinforcement learning. The paper introduces a set of benchmarks built on top of prior work to study this problem. They then introduce a simple approach for tackling this problem that works by augmenting the goal instruction with an action correction instruction.

The paper is  well motivated but the clarity of the presentation can be improved. Examples of instructions with AC and ACN would be helpful. The paper describes three forms of misunderstandings (ambiguity, lack of common ground and instruction correction) but the experiments only seem to consider the third. Further experiments studying the performance and failure modes of the three different types of misunderstandings would be useful.

The experimental setup is very preliminary and could use more baselines (for eg., agent without action correction) and tasks (more complex tasks). I am also a little confused about how the success rates are computed. Is it on a held out set of instructions/corrections? If the objective of this approach is for the usage in online settings like SayCan, it needs to be evaluated in the same way as well. The current sets of experiments do not seem to show that.  In general, after reading the experimental sections, I am not sure what to conclude. It does not seem to indicate anything about its test-time performance. To me, it only seems to show that simple corrections are learnable and negation corrections are harder to learn. Further clarifying and developing the methodology and experimental section would be needed for the final version.

Overall, I think this is an important topic of interest to this community and with modifications would be a good contribution to the workshop program.

---

### Decision · Program_Chairs · 2022-10-21

Accept